# Extra-Heavy Oil Aquathermolysis Using Nickel-Based Catalyst: Some Aspects of In-Situ Transformation of Catalyst Precursor

**Alexey V. Vakhin [1],\*, Firdavs A. Aliev [1] , Irek I. Mukhamatdinov [1] , Sergey A. Sitnov [1], Sergey I. Kudryashov [2], Igor S. Afanasiev [2], Oleg V. Petrashov [2] and Danis K. Nurgaliev [1]**

[1] Institute of Geology and Petroleum Technologies, Kazan Federal University, Kremlyovskaya Str. 18, 420008 Kazan, Russia; firdavsaliev1@gmail.com (F.A.A.); IIMuhamatdinov@gmail.com (I.I.M.); sers11@mail.ru (S.A.S.); Danis.Nourgaliev@kpfu.ru (D.K.N.)

[2] JSC Zarubezhneft, Armiyansky per., 9/1/1, Bld.1, 101990 Moscow, Russia; nestro@nestro.ru (S.I.K.); iafanasiev@nestro.ru (I.S.A.); OPetrashov@nestro.ru (O.V.P.)

\* Correspondence: vahin-a_v@mail.ru

**Abstract:** In the present work, we studied the catalytic performance of an oil-soluble nickel-based catalyst during aquathermolysis of oil-saturated crushed cores from Boca de Jaruco extra-heavy oil field. The decomposition of nickel tallate and some aspects of in-situ transformation of the given catalyst precursor under the steam injection conditions were investigated in a high-pressure batch reactor using XRD and SEM analysis methods. The changes in physical and chemical properties of core extracts after the catalytic aquathermolysis process with various duration were studied using gas chromatography for analyzing gas products, SARA analysis, GC-MS of saturated and aromatic fractions, FT-IR spectrometer, elemental analysis, and matrix-activated laser desorption/ionization (MALDI). The results showed that nickel tallate in the presence of oil-saturated crushed core under the injection of steam at 300 °C transforms mainly into nonstoichiometric forms of nickel sulfide. According to the SEM images, the size of nickel sulfide particles was in the range of 80–100 nm. The behavior of main catalytic aquathermolysis gas products such as $CH_4$, $CO_2$, $H_2S$, and $H_2$ depending on the duration of the process was analyzed. The catalytic upgrading at 300 °C provided decrease in the content of resins and asphaltenes, and increase in saturated hydrocarbon content. Moreover, the content of low-molecular alkanes, which were not detected before the catalytic aquathermolysis process, dramatically increased in saturates fraction after catalytic aquathermolysis reactions. In addition, the aromatics hydrocarbons saturated with high molecular weight polycyclic aromatic compounds—isomers of benzo(a)fluorine, which were initially concentrated in resins and asphaltenes. Nickel sulfide showed a good performance in desulfurization of high-molecular components of extra-heavy oil. The cracking of the weak C–S bonds, which mainly concentrated in resins and asphaltenes, ring-opening reactions, detachment of alkyl substitutes from asphaltenes and inhibition of polymerization reactions in the presence of catalytic complex reduced the average molecular mass of resins (from 871.7 to 523.3 a.m.u.) and asphaltenes (from 1572.7 to 1072.3 a.m.u.). Thus, nickel tallate is a promising catalyst to promote the in-situ upgrading of extra-heavy oil during steam injection techniques.

**Keywords:** heavy oil; in situ upgrading; nickel; aquathermolysis; catalyst; transition metals; SARA-analysis





## 1. Introduction

Conventional hydrocarbon resources are becoming insufficient to supply the ever-increasing demand of steadily growing human energy consumption [1]. The share of renewable energy is still progressing very slow and facing sustainability and storage issues. Hence, hydrocarbon will remain the main source of energy for future decades and unconventional oil reserves, such as heavy and extra-heavy oil, oil shale, tar sands, and bitumen are promising alternative energy sources due to the large share of proven reserves.

However, the high viscosity, density, significant content of high-molecular components (asphaltenes and resins), heteroatoms (S, O, N), low H/C, and various metal complexes such as vanadyl and porphyrin makes the production, transportation, and refinery of such hydrocarbons challenging and unable to recover by conventional methods. Thermal enhanced oil recovery methods (Cyclic steam stimulation (CSS), Steam-assisted gravity drainage (SAGD) and In-situ combustion (ISC)) are widely used to extract heavy and extra-heavy oil. Although, the ISC is considered as the low cost mean to deliver the heat into the reservoir formations, the industrial scale application of this method is limited. The reason is probably due to the difficulty in controlling the whole process and particularly the combustion front. Steam-based recovery techniques are attractive and widely applied in industrial-scale production of heavy and extra-heavy oil, as well as natural bitumen [2]. However, large volumes of water are needed to generate steam. The viscosity of heavy oil after steam treatment temporary decreases, which is not favor in terms of transportation and refinery of heavy oil [3]. The feasibility of steam recovery techniques depends on the steam-to-oil-ratio (SOR) and the market price for crude oil [4]. Without a doubt, the SOR of steam recovery methods needs to be decreased and the efficiency of steam injection has to be improved. Injection of steam into the heavy oil formation involves physical, as well as chemical consequences. The physical aspect of steam injection is a well-known drive mechanisms: high temperature and pressure, emulsification, wettability and viscosity reduction. On the other hand, the overall set of chemical interactions that occur between the heavy oil, rock minerals and high temperature water or steam independent from the phase behavior are called Aquathermolysis reactions. As far back as 1982, the aquathermolysis has been evidenced as a distinct process by Hyne et al. [5]. The content of high molecular components such as resins and asphaltenes in heavy and extra-heavy oil mostly prevails 50 wt.% [3,6]. The efficiency of upgrading and irreversible viscosity reduction depends on the cracking of these species with further hydrogenation. The decomposition of asphaltenes as a representative of the heaviest fraction of heavy and extra-heavy oil can be initiated early at 120 °C through the cleavage of the weakest C–S and S–S bonds. The destruction products of asphaltenes are attacked by hydrogen, which inhibits polymerization reactions. In this regard, the source of hydrogen is crucial during in-situ aquathermolysis of heavy and extra-heavy oil. As the injection of $H_2$ is not safe, many attempts have been made to generate the hydrogen in-situ [7–11]. In addition, water dissociation and water-shift-gas reactions may produce $H_2$ [5]. However, the produced amount of $H_2$ is probably insufficient for catalytic upgrading of heavy and extra-heavy oil. Therefore, additional hydrogen donors are introduced in order to supply the hydrogen during catalytic upgrading processes. Moreover, the overall chemical reactions have to be accelerated in order to generate more $H_2$ and intensify in-reservoir upgrading processes. Many researchers have been developing various catalysts and tested hydrogen donor solvents, which could promote the aquathermolysis reactions and further assist in downhole upgrading of heavy oil [12–18]. In this study, we used a Solvent as a hydrogen donor, which composed of mainly cycloalkanes. The composition of the given solvent is thoroughly discussed in our earlier study [19]. The combination of catalysts with hydrogen donors (catalytic complex) is crucial to enhance the heavy oil production and consequently decrease the SOR.

In general, four types of aquathermolysis catalysts are known: mineral, water-soluble, dispersed and oil-soluble catalysts [20]. Fan et al. reported that the natural minerals have catalytic effect on aquathermolysis of heavy oil. He claims that adding 10 wt.% mineral to the system increases the share of light hydrocarbons and decreases the content of resins and asphaltenes [21]. However, the viscosity reduction degree in the presence of only mineral was only 25% [21]. Thus, the steam treatment of heavy oil in the presence of only rock minerals is not enough to achieve required upgrading degree. Another type of catalysts, which are the most thoroughly studied ones, are water-soluble salts of transition metals such as $FeSO_4$, $RuCl_3$, $NiSO_4$, $VOSO_4$, etc. [22]. Such catalysts were successfully used in the early works of Clark et al. [23–25]. Later on, the effect of various water-soluble catalysts

on aquathermolysis of Liaohe extra-heavy oil was reported [26,27]. The authors draw our attention on the catalytic cracking of C–S, C–N, and C–O bonds initiating and promoting overall aquathermolysis reactions that lead to the decrease in the content of resins and asphaltenes. Hence, the viscosity after catalytic aquathermolysis (at 240 °C) of extra-heavy oil from Liaohe field for 24 h was reduced up to 75%. Fan et al. tested combination of water-soluble catalysts with KOH/NaOH in heavy oils from 10 different fields, where up to 93% of viscosity reduction degree was achieved [28]. The experimental results obtained by Zhong and co-workers also affirm 90% viscosity reduction after aquathermolysis of extra-heavy oil from Liaohe in the presence of Fe(II) salts and tetralin [3]. A key problem with much of the literature on industrial-scale application of water-soluble catalysts is a poor contact of metal surfaces with oil phase. The distribution of such metal ions is limited with the interface of water-oil system due to insolubility in hydrocarbons. The high-molecular components of crude oil, which are deterministic factor of oil mobility, remain unreacted as they concentrate in low-permeable reservoir rocks and hence, the water-soluble catalysts are often impractical due to their activity. In this regard, nanocatalysts are attracting considerable interest due to the huge surface area. Dispersed catalysts based on transition metals, oxide, sulfide, carbide, phosphide nanoparticles and metalized solid acids were discussed in the review article [29]. Pevnevo and co-workers carried out comparison study on the aquathermolysis of fuel oil in the presence and absence of a tungsten carbide/nickel-chromium. The presence of catalysts promoted the destructive hydrogenation reactions of heavy fractions and resulted to the increase in the content of $C_9$–$C_{17}$ n-alkanes [30]. Other authors imply that La/MCM-41/-$Al_2O_3$ provides intense desulfurization (up to 40%) during cracking of vacuum gasoil [31]. An alternative approach of obtaining molybdenum oxide nanocatalyst from inverted emulsion of aqueous paramolybdate ammonium solution was proposed by Kadieva et al. [32]. A major drawback of using heterogeneous nanoparticles is their dispersity and stabilization in oil phase. Several issues arise during injection of nanoparticles through wellbore under high pressure and further flow through porous media. Moreover, the cost of industrial synthesis of nanoparticles is very high.

There is a vast amount of literature on oil-soluble catalysts for the aquathermolysis of heavy oil [6,32,33]. The formation of such oil-soluble catalysts from the precursors is the best solution in terms of catalysts' delivery to the reservoir formations. Moreover, the solubility of such catalysts in oil phase increases the efficiency of heavy oil aquathermolysis catalysis [18,34,35]. In its turn, solubility strongly depends on the organic ligand. Carboxylates, oleates and naphthenic acids are considered as the most applied ligands [33,34]. The hydrogen donating ability of dimethylbenzenesulfonic ligands were reported in [35]. Suwaid et al. used stearic acid as a ligand to synthesize the oil-soluble catalyst precursor for aquathermolysis of heavy oil from Ashal'cha field (Tatarstan). Thus, screening criteria for ligands should be their solubility in crude oil, decomposition temperature of ligands and their accessibility. In this study, we used distilled tall oil as a green and cheap ligand for catalyst precursor. Tall oil is a by-product of wood pulp processing composed of various fatty acids depending on the type of woods. A common accepted mean to describe the quality of tall oil is its acidic number [36]. The acidic number for distilled tall oil used in this study is equal to 182 mg KOH/g. Previously, we have conducted a series of studies on oil-soluble catalysts based on transition metal tallates to optimize the composition of catalytic complex [6,37]. It was revealed that nickel tallate in combination with hydrogen donor (Solvent) performed the highest activity in terms of conversion of asphaltenes and resins into light hydrocarbons [15]. The increase in the content of low-molecular alkanes after the catalytic upgrading of extra-heavy oil was presented in [38]. In this study, nickel-based oil-soluble catalyst is deeply studied by multiple methods to reveal their catalytic activity duration up to 96 h, in-situ transformation mechanism, and possible catalytic mechanism in aquathermolysis of extra-heavy oil.

## 2. Results and Discussions

### 2.1. Activation of Catalyst Precursors

XRD patterns of isolated Ni-based catalysts from the products of extra-heavy oil aquathermolysis carried out for 1–12 h are provided in Figure 1.

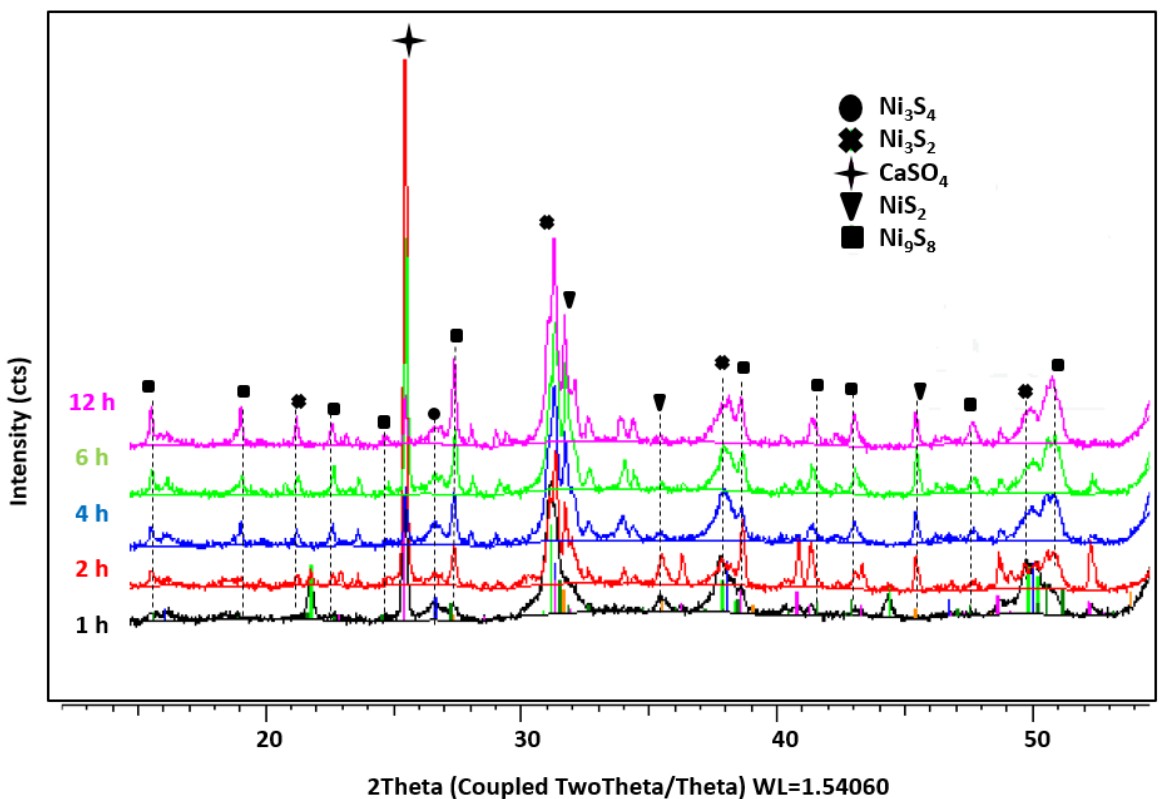

**Figure 1.** XRD patterns of isolated Ni-based catalyst samples after thermo-catalytic treatment for different duration.

After hydrothermal treatment of extra-heavy oil at 300 °C in the presence of nickel tallate, nickel was found mainly in the forms of nickel sulfide. The catalytic role of nickel sulfide in upgrading of heavy oil and its desulfurization is all accepted fact. Nonstoichiometric forms of nickel sulfide and their relative content, depending on the aquathermolysis duration, are listed in Table 1.

Nonstoichiometric forms of nickel sulfide are $NiS_2$, $Ni_9S_8$, $Ni_3S_4$, and $Ni_3S_2$. The composition of nickel sulfide changes with the reaction time. The $Ni_3S_2$ is not detected in the composition of samples, which were treated more than 2 h. The relative content of sulfur sharply increases from 20.3% (1 h) to 50.6% (2 h) and remains at levels of 51–53%.

The SEM image and compositions of catalyst surface isolated from extra-heavy oil after aquathermolysis is presented in Figure 2. According to the image, the size of catalyst particles was in the range of 80–100 nm. However, aggregates of micrometer size can be observed, which probably correspond to coke-like compounds such as carbenes and carboids. They form after the cracking of asphaltene molecules due to the loss of alkyl substitutes and functional groups in asphaltenes. It has to be noted that the sizes of catalyst particles isolated after different aquathermolytic treatment time were within the same range.

**Table 1.** The phases and contents of nickel-based catalyst after hydrothermal exposure.

| Aquathermolysis Duration, Hour(s) | Phases | Content, wt.% | Content of Ni and S, Rel.% | |
|---|---|---|---|---|
| | | | Ni | S |
| 1 | $NiS_2$ | 5 | 79.8 | 20.2 |
| | $Ni_3S_4$ | 20 | | |
| | $Ni_3S_2$ | 22 | | |
| | $Ni_9S_8$ | 24 | | |
| | $CaSO_4$ | 28 | | |
| 2 | $NiS_2$ | 6 | 49.4 | 50.6 |
| | $Ni_3S_2$ | 8 | | |
| | $Ni_3S_4$ | 9 | | |
| | $Ni_9S_8$ | 20 | | |
| | $CaSO_4$ | 58 | | |
| 4 | $K_2Ca(CO_3)_2$ | 4 | 48.2 | 51.8 |
| | $NiS_2$ | 8 | | |
| | $Ni_3S_2$ | 0 | | |
| | $CaSO_4$ | 10 | | |
| | $Ni_3S_4$ | 25 | | |
| | $Ni_9S_8$ | 53 | | |
| 6 | $K_2Ca(CO_3)_2$ | 4 | 48.4 | 51.6 |
| | $NiS_2$ | 6 | | |
| | $Ni_3S_4$ | 21 | | |
| | $Ni_3S_2$ | 0 | | |
| | $CaSO_4$ | 24 | | |
| | $Ni_9S_8$ | 45 | | |
| 12 | $NiS_2$ | 3 | 47.0 | 53.0 |
| | $K_2Ca(CO_3)_2$ | 4 | | |
| | $CaSO_4$ | 7 | | |
| | $Ni_3S_2$ | 0 | | |
| | $Ni_3S_4$ | 24 | | |
| | $Ni_9S_8$ | 62 | | |

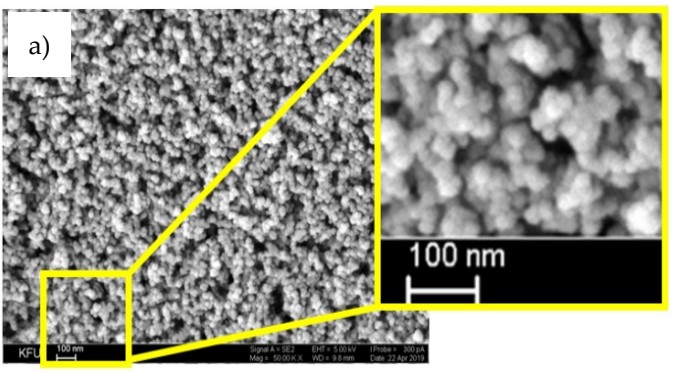
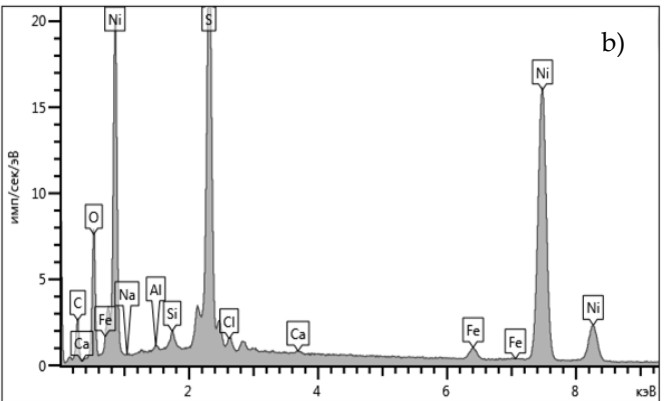

**Figure 2.** SEM image of catalyst particles (**a**) and their surface composition (**b**).

The distribution of elements in the catalysts isolated after thermo-catalytic treatment for 12 h are presented in Figure 3. The results indicate the ambiguous behavior in the weight compositions of Ni, C, O, S elements. During various thermo-catalytic treatment duration, the active form of the catalyst was observed in the form of nickel sulfide. The adsorption of coke-like compounds on the catalyst nanoparticles results to the increase in the content of carbon element. The content of Na and Al metals as well as non-metals such as O, Si, and Cl, which exist in the composition of rocks, were increased.

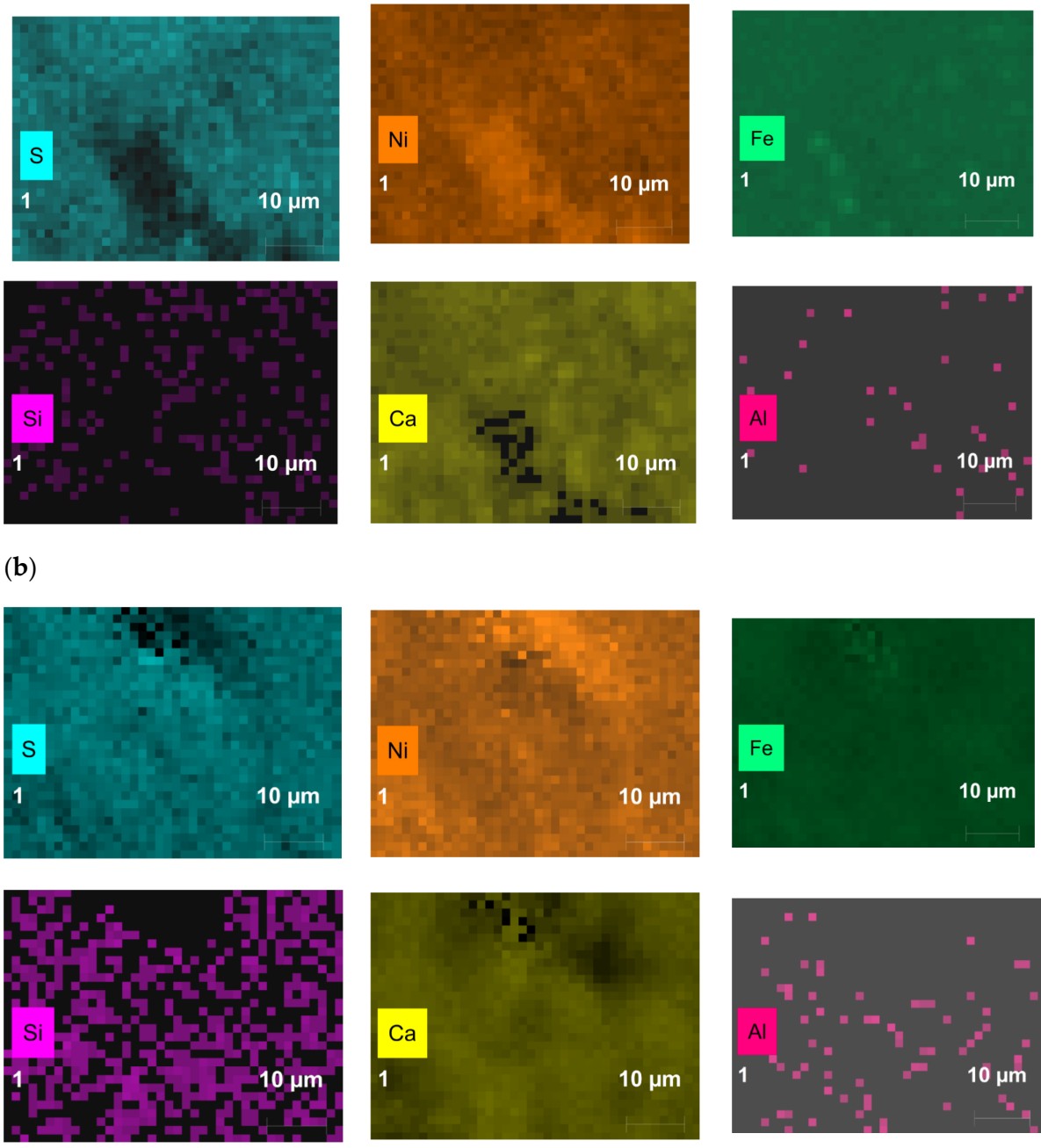

**Figure 3.** *Cont.*

(**c**)

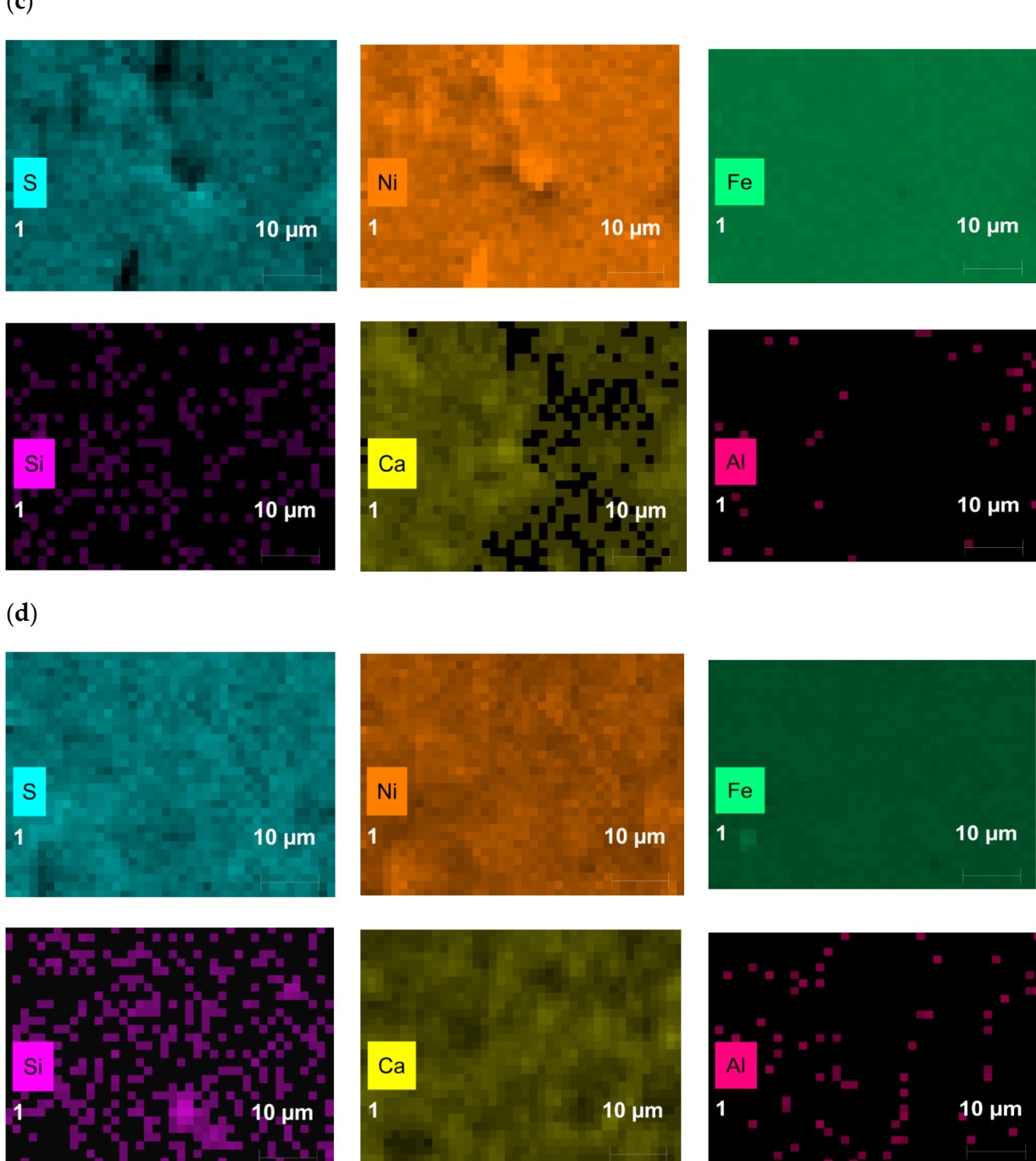

(**d**)

**Figure 3.** *Cont.*

(**e**)

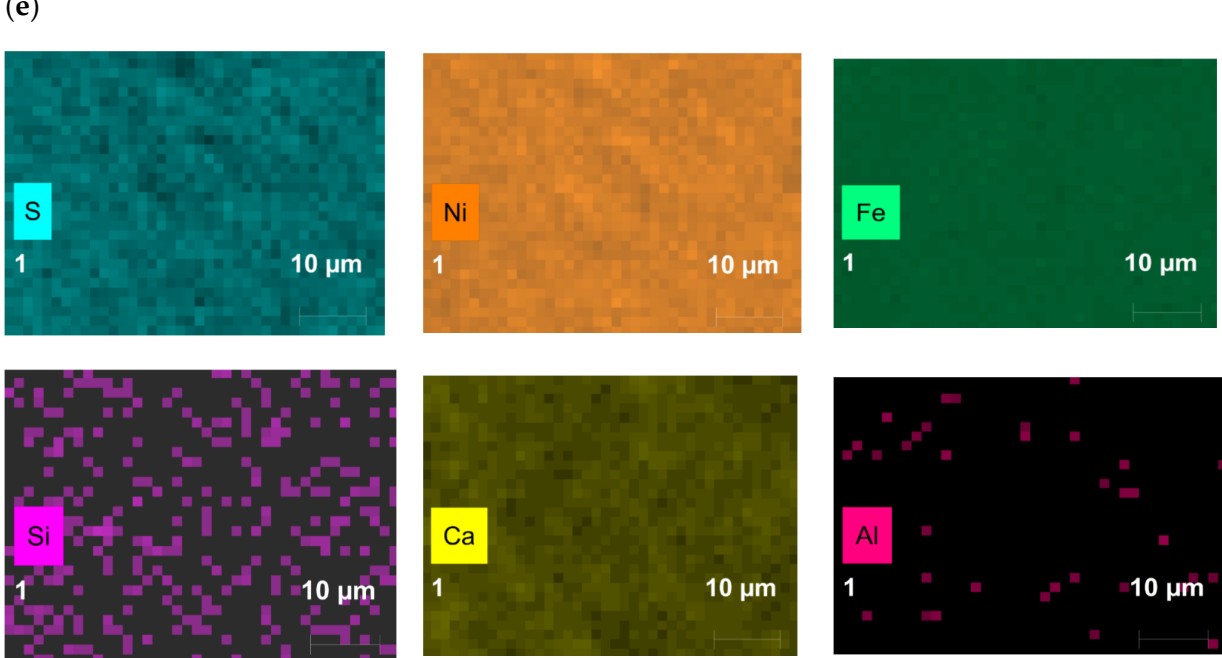

**Figure 3.** The distribution of elements in the catalysts isolated after thermo-catalytic treatment for (**a**) 1 h; (**b**) 2 h; (**c**) 4 h; (**d**) 6 h; (**e**) 12 h.

### 2.2. Gas Phase Products of Catalytic and Non-Catalytic Aquathermolysis

Table 2 shows the main gas components of catalytic and non-catalytic aquathermolysis (48–96 h) products of oil-saturated crushed cores. They are seen to be precisely the same chemical compounds ($CH_4$, $CO_2$, $H_2S$ and $H_2$) as those generated from aquathermolysis of organosulphur model compounds—thiolane and thiophene, which confirm the cracking of C–S bonds [20]. However, the relative content of these gaseous compounds are lower since the amount of organosulphur species in crushed core extracts is very low in contrast to the pure model of organosulphur compounds. Hyne et al. reported that generation of gases after aquathermolysis of thiolane is much higher than thiophene [39]. Hence, the production of gases depends not only on the amount of organosulphur compounds in oil composition, but also on the chemical nature (i.e., reactivity) of organic compounds that contain sulfur. The gas chromatography results after catalytic aquathermolysis of crushed core samples show the decrease for hydrogen amount with the increase in duration of aquathermolysis process (Table 2). This can be explained by involving hydrogen in the in-reservoir hydrocracking process in the presence of water phase at relatively low temperatures and in stability of alkyl radicals. The content of low-molecular-weight alkanes ($C_1$–$C_5$) increases with the duration of catalytic aquathermolysis process, while the content of unsaturated hydrocarbons decreases. Here, the role of water in adding hydrogen to the extra-heavy oil becomes significant in long-time aquathermolytic experiments. The hydrogen donating capacity of water in aquathermolytic reactions was studied and proved by several authors [40,41].

The significant production of $H_2S$ at temperatures above 240 °C during steam recovery of heavy oils is well established. The evolvement of hydrogen sulphide gases during catalytic aquathermolysis is mainly the result of thermal and hydrolysis of sulphur-containing moieties in asphaltene molecules. Besides, the thermal decomposition of sulfides and thiols, which are promoted in the presence of carbonate rocks and high-temperature hydrolysis of organosulphur compounds, lead to the formation of hydrogen sulfides as well. Carbon dioxide is another main gas component detected by Gas Chromatography (GC) after catalytic aquathermolysis reactions. The proposed source of $CO_2$ generation is Water-shift-gas reaction (WSGR). The carbon oxide and dioxide production is also possible during destruc-

tion of mono- and dioxide-containing groups and sulfoxides in the oil composition. The carbonates are also considered the possible source of $CO_2$ at the steam treatment conditions. It is interesting to compare the amount of released $CO_2$ after 96 h catalytic aquathermolysis of oil-saturated crushed core (54.65 vol.%) and extra-heavy oil (26.95 vol.%). Independent from the source of $CO_2$, its generation assists in reduction of extra-heavy oil viscosity.

**Table 2.** Main gas components after catalytic and non-catalytic aquathermolysis of oil-saturated crushed cores.

| Experimental Conditions | | | With Catalyst | | | Without Catalyst |
|---|---|---|---|---|---|---|
| **Duration, Hours** | | | **48** | **72** | **96** | **96** |
| Composition of gaseous products, wt.% | Saturated hydrocarbons | $H_2$ | 0.54 | 0.71 | - | 0.42 |
| | | $CO_2$ | 55.71 | 58.61 | 54.65 | 45.17 |
| | | $H_2S$ | 17.09 | 10.38 | 13.14 | 14.70 |
| | | $CH_4$ | 13.26 | 14.38 | 15.82 | 16.54 |
| | | $C_2H_6$ | 7.21 | 9.28 | 9.20 | 9.54 |
| | | $C_3H_8$ | - | - | - | 6.85 |
| | | normal (n)-$C_4H_{10}$ | 1.49 | 2.02 | 2.14 | 2.11 |
| | | iso (i)-$C_4H_{10}$ | 0.64 | 0.91 | 0.94 | 1.05 |
| | | n-$C_5H_{12}$ | 0.52 | 0.65 | 0.72 | 0.64 |
| | | neo-$C_5H_{12}$ | 0.21 | 0.22 | 0.24 | - |
| | | i-$C_5H_{12}$ | 0.35 | 0.50 | 0.54 | 0.57 |
| | | n-$C_6H_{14}$ | 0.27 | 0.25 | 0.26 | 0.24 |
| | | i-$C_6H_{14}$ | 0.18 | 0.20 | 0.21 | 0.22 |
| | | Isomers $C_7$–$C_8$ | 0.42 | 0.32 | 0.15 | 0.38 |
| | Unsaturated hydrocarbons | $C_2H_4$ | 0.09 | - | - | - |
| | | $C_3H_6$ | 0.40 | - | - | - |
| | | $C_4$–$C_5$ | 0.48 | 0.37 | 0.06 | 0.28 |
| | | $C_6$–$C_7$ | 0.17 | 0.17 | 0.23 | 0.15 |
| | | Sum. | 1.14 | 0.54 | 0.29 | 0.43 |
| Total amount gaseous products, g/100 g of core | | | 0.622 | 0.686 | 0.742 | 0.725 |

### 2.3. SARA Analysis of Core Extracts before and after Catalytic Aquathermolysis

The group composition of core extracts depending on duration of catalytic aquathermolysis are presented in Figure 4. The SARA analysis of core extracts treated less than 48 h were reported in [15]. The time dependence of the catalytic aquathermolysis at 300 °C is obviously observed in destruction of asphaltenes. The content of saturates doubled at relatively short-time treatment periods (6–24 h) due to detachment of alkyl substitutes. The required energy for such process is less. Therefore, the long-time catalytic aquathermolysis is not further increasing the content of saturates. The sharp decrease in the content of asphaltenes was up to 72 h. The further increase in duration of reactions was not effective, as a minimum achievable content was achieved. Asphaltenes are characterized by maximum condensation and the further cracking of high-molecular weight components in the given conditions is impossible. The increase in the content of resins up to 48 h (Figure 4) can be reasoned by joining cracking products of asphaltene molecules. Above 48 h, the asphaltene cracking products no longer join the resin fraction and hence, the content of resins declines due to their chemical destruction. The chemical destruction of resins increases aromatic hydrocarbons. GC-MS results showed new components of aromatic fractions, which were destruction products of resins and asphaltenes. Thus, the

catalytic upgrading of extra-heavy oil is the result of several reactions such as thermal destruction, cyclization, dehydrogenation, hydrolysis and alkylation of aromatic rings by intermediate products. The increase in the content of paraffinic-naphthenic and aromatic hydrocarbons with duration of catalytic aquathermolysis is due to the enhancing the influence of cracking reaction.

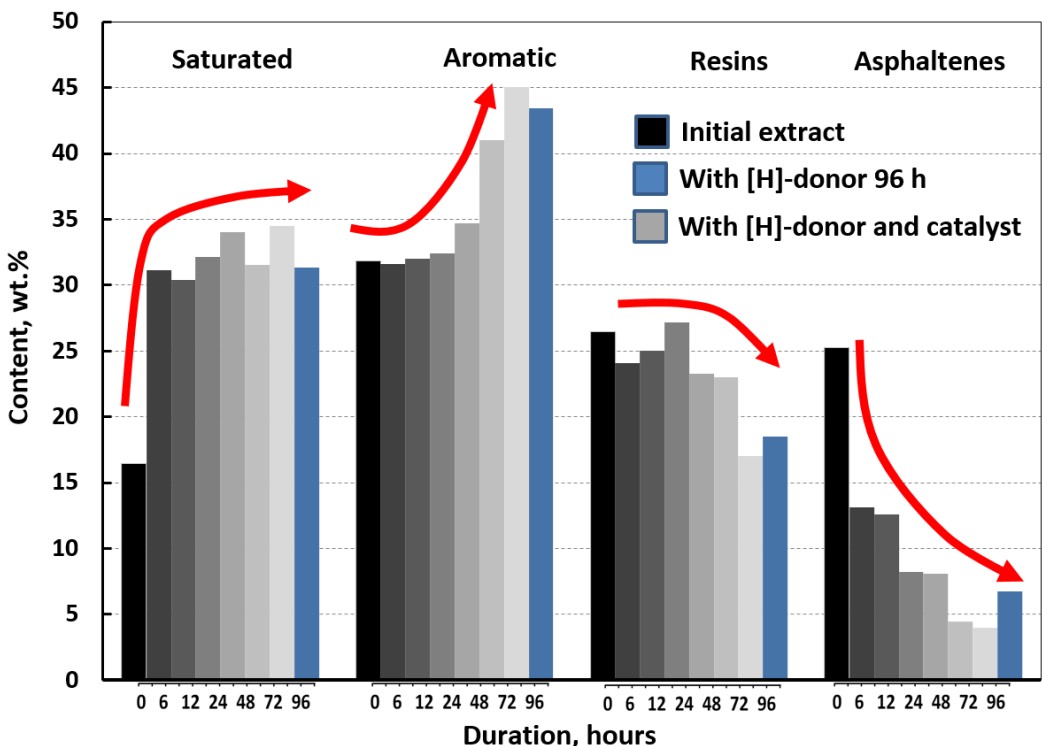

**Figure 4.** SARA analysis of core extracts depending on duration of catalytic aquathermolysis.

### 2.4. GC-MS Analysis of Saturates and Aromatics Hydrocarbons

GC-MS results of saturates and aromatics fractions depending on the duration of catalytic aquathermolysis are presented in Figures 5 and 6, accordingly. The low molecular weight alkanes were not present in the initial core extracts. The content of them dramatically increases after catalytic aquathermolysis reactions. The spectra of newly generated hydrocarbons widens after 96 h treatment mainly due to $C_{13}$–$C_{18}$ alkanes. This indicates the deep conversion of resins and asphaltenes hydrocarbons. Detachment of alkyl substitutes from asphaltenes after thermo-catalytic treatment were reported in several works [42,43]. Such detachment products increase the content of saturated hydrocarbons, which significantly effects the viscosity of extra-heavy oil.

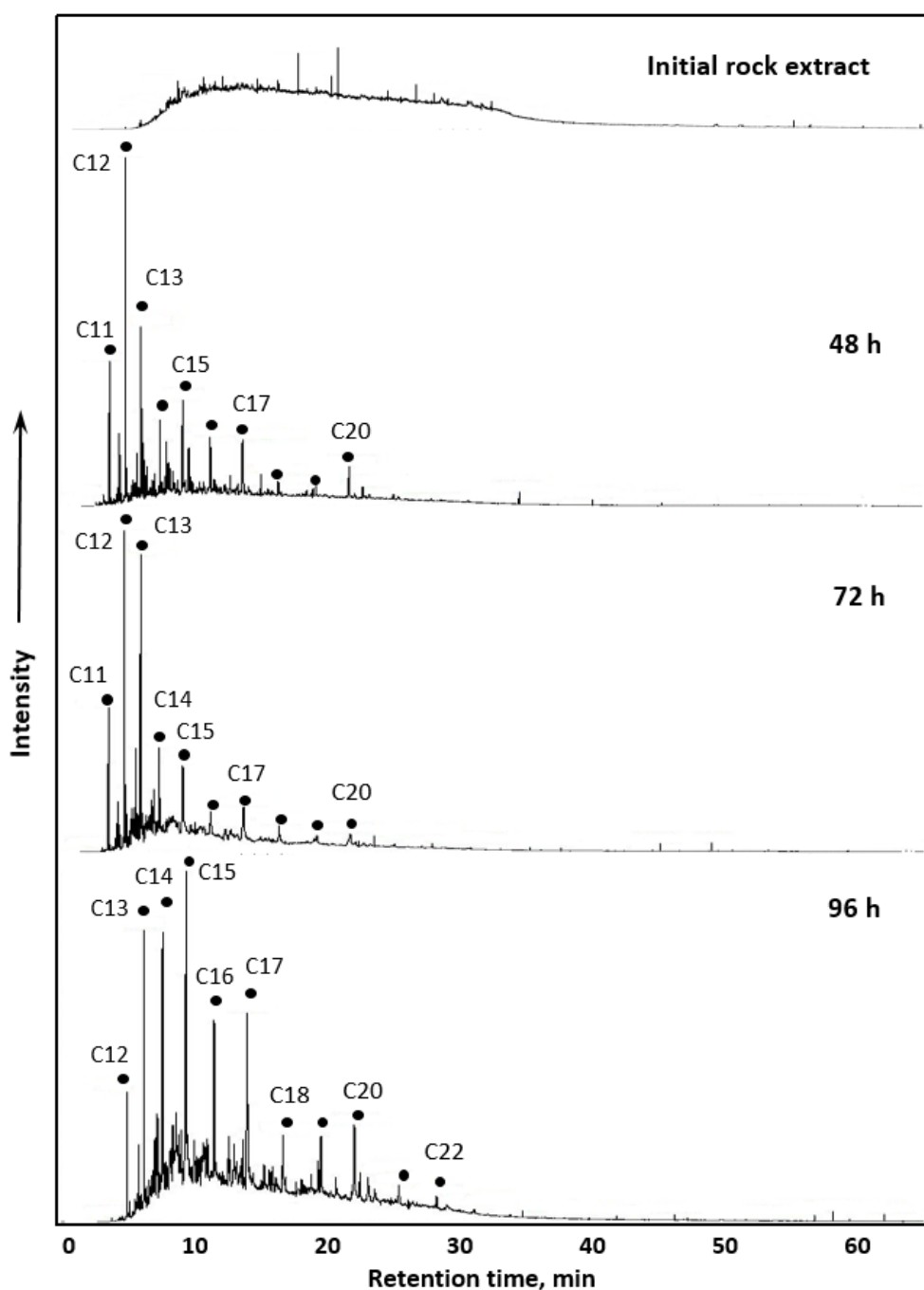

**Figure 5.** Total ion chromatograms of saturated hydrocarbons depending on the duration of catalytic aquathermolysis.

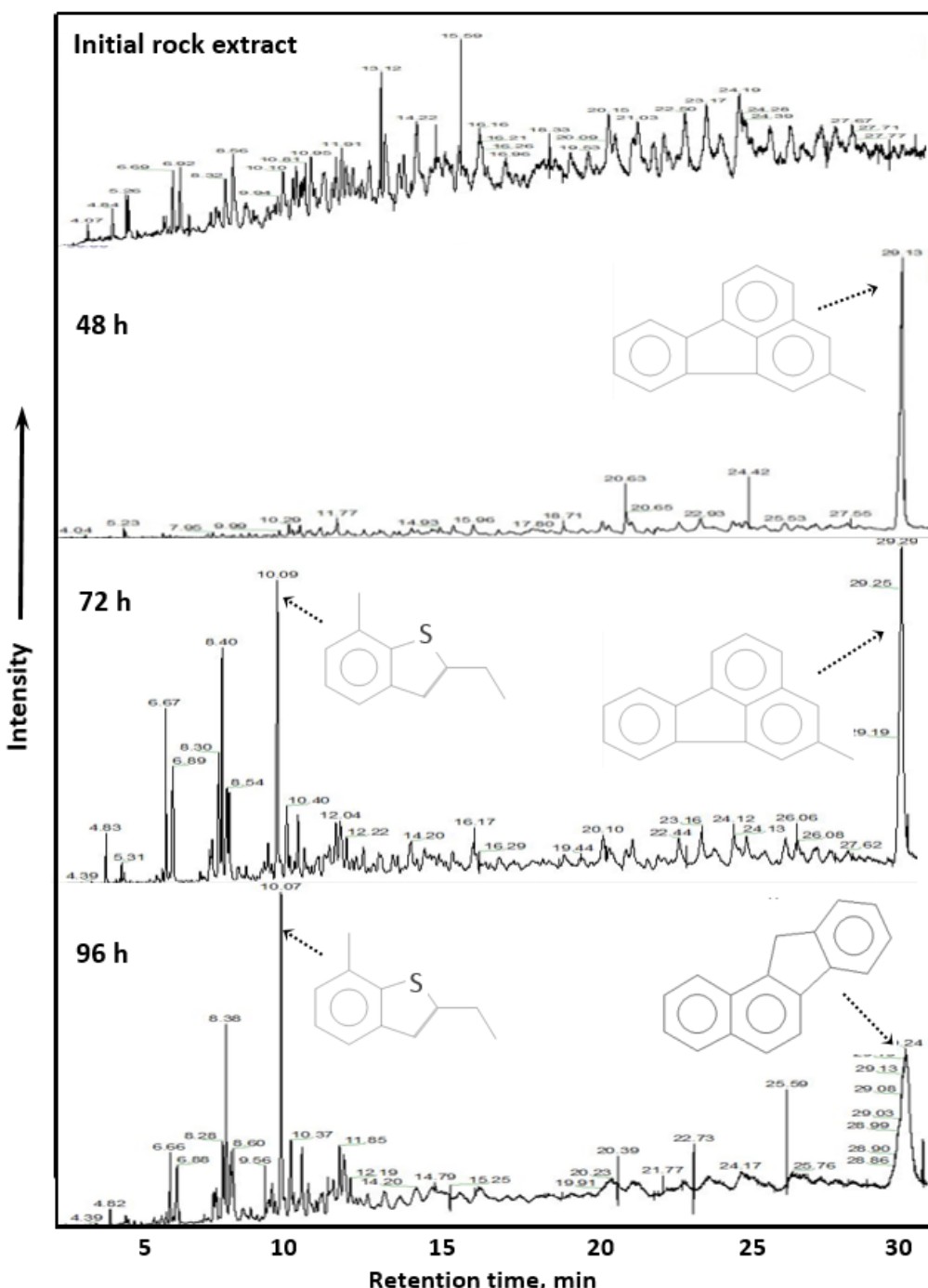

**Figure 6.** Total ion chromatograms of aromatic hydrocarbons depending on the duration of catalytic aquathermolysis.

The aromatics hydrocarbons saturate with high molecular weight polycyclic aromatic compounds such as isomers of benzo(a)fluorine, which are concentrated in resins and asphaltenes. This accord with the results of SARA-analysis, where a significant increase in the content of aromatics (after 48 h) was observed with a decrease in the content of resins and asphaltenes. The similar phenomenon was observed earlier in the products of catalytic aquathermolysis of the given extra-heavy oil for 24 h with prevailing phenanthrenes in aromatic fractions [6].

In Figure 7, we demonstrate the comparison of saturated and aromatic hydrocarbons in the presence and absence of nickel-based oil soluble catalyst. The hydrogen donor presents in both cases. The role of carbonate rock minerals (Figure 7, upper spectra) is small in chemical transformation of resins and asphaltenes, even for long period treatment. The impact of catalyst is not only in promoting the process, but also in deep conversion of high molecular alkyl and aromatic substitutes. The content of $C_{17}$–$C_{20}$ significantly increases in the presence of nickel-based catalyst. The high molecular polycyclic aromatic hydrocarbons, which are probably the destruction products of asphaltene fragments were not detected in the aromatics hydrocarbons without catalyst.

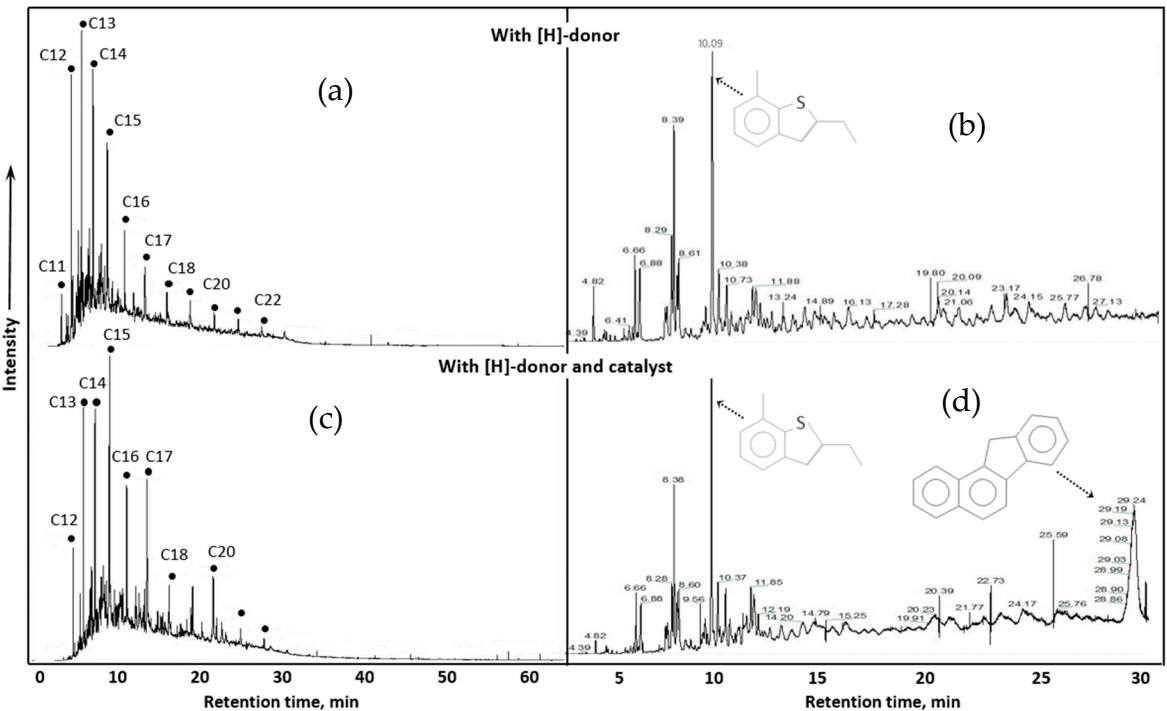

**Figure 7.** Total ion chromatograms of saturated hydrocarbons (**a,c**) and aromatic hydrocarbons (**b,d**) after aquathermolysis in the presence (**c,d**) and absence (**a,b**) of nickel-based catalyst for 96 h.

Table 3 shows the composition of saturated hydrocarbons depending on the duration of catalytic aquathermolysis. There was no linear alkane observed in the composition of saturates of initial crushed core extracts. However, the catalytic upgrading for 48 h leads to the increase in the content of mainly low molecular weight alkanes ($C_{11}$–$C_{15}$), which stand for 85% of total normal alkanes ($C_{11}$–$C_{22}$). The further increase (72 h) in duration of catalytic aquathermolysis reaction results to the noticeable increase in the content of $C_{13}$ and hence the sum of $C_{11}$–$C_{15}$ is increased. Catalytic upgrading for 96 h exhibits reduction in the content of $C_{11}$–$C_{15}$, while the content of relatively high molecule weight components (above $C_{16}$) increases. These results show that some long peripheral chains in resins and asphaltenes were cracked in the presence of nickel-based catalytic complex due to ring opening, hydrothermal cracking at lower temperatures and hydrogenation reactions.

**Table 3.** Relative content of normal alkanes in saturated hydrocarbon fractions after catalytic upgrading of crushed cores for different durations.

| Alkanes | Initial Core Extract | After catalytic Upgrading, % | | |
|:---:|:---:|:---:|:---:|:---:|
| | | 48 h | 72 h | 96 h |
| $C_{11}$ | | 15.38 | 15.43 | 0.41 |
| $C_{12}$ | | 36.54 | 33.95 | 9.13 |
| $C_{13}$ | | 17,63 | 29.63 | 18.26 |
| $C_{14}$ | | 6.41 | 9.26 | 17.24 |
| $C_{15}$ | | 9.62 | 6.17 | 20.28 |
| $C_{16}$ | | 4.81 | 0.93 | 10.75 |
| $C_{17}$ | Not detected | 4.81 | 2.78 | 11.36 |
| $C_{18}$ | | 0.64 | 0.93 | 4.06 |
| $C_{19}$ | | 0.64 | 0.31 | 3.45 |
| $C_{20}$ | | 2.88 | 0.62 | 4.06 |
| $C_{21}$ | | 0.32 | 0.00 | 0.61 |
| $C_{22}$ | | 0.32 | 0.00 | 0.41 |
| $C_{11}$–$C_{15}$ | | 85.58 | 94.44 | 65.31 |

### 2.5. FT-IR Spectroscopy Results

The spectrometric indices of crushed core extracts before and after the catalytic upgrading depending on the duration of the process are summarized in Table 4 and their spectra are presented in Figure 8. The results exhibit a slight decrease in aromaticity index ($C_1$) with increasing the duration of the catalytic upgrading due to sciss of peripheral alkyl radicals and long hydrocarbon chains from resins and asphaltenes. The previous statement is true if only aliphatic indices ($C_4$) are increasing with decreasing aromaticity indices. This is explained by production of low molecular weight alkanes after catalytic upgrading.

**Table 4.** FT-IR spectrometric indices of crushed core extracts.

| Experimental Conditions | | Spectral Coefficients | | | | |
|:---:|:---:|:---:|:---:|:---:|:---:|:---:|
| | | *$C_1$ | *$C_2$ | *$C_3$ | *$C_4$ | *$C_5$ |
| | Initial core extracts | 0.33 | 0.12 | 0.58 | 7.38 | 0.20 |
| 48 h | | 0.39 | 0.11 | 0.58 | 7.00 | 0.15 |
| 72 h | Hydrothermal treatment in the presence of catalyst and [H]-donor | 0.37 | 0.10 | 0.58 | 7.32 | 0.15 |
| 96 h | | 0.34 | 0.11 | 0.57 | 7.63 | 0.13 |
| | Without catalyst | 0.33 | 0.04 | 0.57 | 7.52 | 0.14 |

*$C_1$ = $D_{1600}/D_{720}$ (aromaticity), *$C_2$ = $D_{1710}/D_{1465}$ (oxidation), *$C_3$ = $D_{1380}/D_{1465}$ (branching), *$C_4$ = $(D_{720} + D_{1380})/D_{1600}$ (aliphatic) and *$C_5$ = $D_{1030}/D_{1465}$ (sulfurization).

The insignificant changes in oxidation ($C_2$) and branching ($C_3$) indices indicate the conduction of weak oxidative reactions initiated by water during catalytic aquathermolysis processes. Sulfurization index ($C_5$) is decreasing with time due to reduction of sulfoxides to sulfides and hydrogen sulfides, which is consistent with gas chromatography results. The index of aliphatic ($C_4$) is a maximum (7.63) after catalytic aquathermolysis for 96 h due to the minimum aromaticity (0.34) achieved in this sample. This indicates the conduction of bond destruction reactions and ring opening of polycyclic aromatic hydrocarbons.

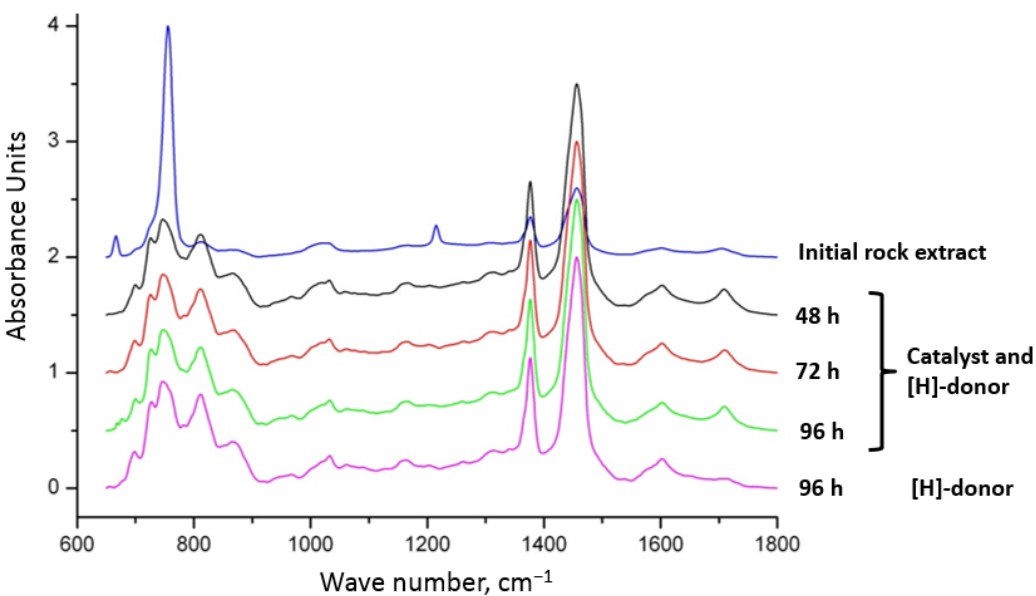

**Figure 8.** FT-IR of core extracts depending on the duration of catalytic hydrothermal treatment at 300 °C.

### 2.6. Elemental Analysis Results

The results of elemental composition of core extracts and their SARA fractions before and after catalytic upgrading are presented in Table 5. The catalytic upgrading has a significant influence on heavy fractions of oil. Thermodynamically, C-S bond requires the minimum energy equal to 272 kJ/mole to be cracked [44]. It is much lower than C–C and even C–N bond energies. Hence, cracking of C–S bonds is much easier during hydrothermal treatment processes. The significant changes are observed in the samples, which are catalytically upgraded for 96 h rather than treating without catalyst. In the latter, the content of sulfur and oxygen is increasing. The influence of catalytic complex (oil-soluble nickel and hydrogen donor) on generation of hydrogen is increasing with duration of catalytic aquathermolysis reactions. This explains the increase of hydrogen to carbon ratio.

**Table 5.** Elemental composition of core extracts and SARA fractions.

| Experimental Conditions | Object | | Elemental Composition, wt.% | | | | | $H/C_{at}$ |
|---|---|---|---|---|---|---|---|---|
| | | | C | H | N | S | O | |
| Initial | Core extracts | | 83.9 | 9.1 | 0.4 | 2.6 | 4.0 | 1.30 |
| | SARA-fractions | S | 84.5 | 12.8 | 0.1 | 1.0 | 1.6 | 1.82 |
| | | A | 80.3 | 9.9 | 0.1 | 4.0 | 5.7 | 1.47 |
| | | R | 79.7 | 8.4 | 0.6 | 4.7 | 6.5 | 1.26 |
| | | A | 73.3 | 8.0 | 0.8 | 7.7 | 10.2 | 1.31 |
| 48 | Core extracts | | 83.1 | 10.3 | 0.4 | 2.3 | 4.0 | 1.49 |
| | SARA-fractions | S | 84.5 | 12.8 | 0.1 | 1.1 | 1.6 | 1.82 |
| | | A | 82.1 | 9.2 | 0.1 | 3.5 | 5.0 | 1.35 |
| | | R | 75.2 | 8.6 | 0.8 | 6.4 | 8.9 | 1.38 |
| | | A | 77.7 | 6.3 | 1.3 | 6.3 | 8.4 | 0.98 |
| 72 | Core extracts | | 82.9 | 10.5 | 0.3 | 2.1 | 4.2 | 1.52 |
| | SARA-fractions | S | 85.3 | 13.0 | 0.1 | 0.6 | 1.0 | 1.83 |
| | | A | 81.9 | 9.2 | 0.1 | 3.6 | 5.2 | 1.35 |
| | | R | 78.1 | 8.8 | 1.0 | 5.1 | 7.0 | 1.35 |
| | | A | 78.6 | 6.6 | 1.2 | 5.9 | 7.7 | 1.01 |
| 96 | Core extracts | | 83.1 | 10.7 | 0.4 | 1.9 | 3.9 | 1.55 |
| | SARA-fractions | S | 84.5 | 12.8 | 0.1 | 1.0 | 1.6 | 1.82 |
| | | A | 80.3 | 9.9 | 0.1 | 4.0 | 5.7 | 1.47 |
| | | R | 79.7 | 8.4 | 0.6 | 4.7 | 6.5 | 1.26 |
| | | A | 83.3 | 8.0 | 0.8 | 5.8 | 9.5 | 0.99 |
| 96 without catalyst | Core extracts | | 82.4 | 10.1 | 0.3 | 2.9 | 4.3 | 1.47 |
| | SARA-fractions | S | 81.5 | 12.6 | 0.1 | 2.3 | 3.5 | 1.86 |
| | | A | 81.4 | 8.8 | 0.1 | 4.0 | 5.7 | 1.30 |
| | | R | 79.9 | 8.2 | 1.3 | 4.5 | 6.2 | 1.24 |
| | | A | 80.6 | 6.8 | 1.2 | 4.9 | 6.5 | 1.01 |

*2.7. Matrix-Activated Laser Desorption/Ionization (MALDI) Analysis Results*

Figure 9 shows the matrix-activated laser desorption/ionization (MALDI) spectra of resins and asphaltenes of core extracts before and after the catalytic aquathermolysis for 48–92 h. A decrease in average molecular mass of resins and asphaltenes was observed. The average molecular mass of resins reduces from 871.7 to 523.3 a.m.u. For asphaltenes, the average molecular mass reduces from 1572.7 to 1072.3 a.m.u. These reductions in average molecular masses are mainly due to detachment of alkyl substitutes in both saturated and unsaturated compounds. The presence of hydrogen donor inhibits the polymerization of free radicals. Thus, the thermo-catalytic upgrading provides not only decrease in the content of resins and asphaltenes according to SARA results, but also reduces their molecular masses. The last is crucial in terms of mobility through porous media. The achieved MALDI results after catalytic upgrading processes are consistent with the results reported by other researchers [45,46].

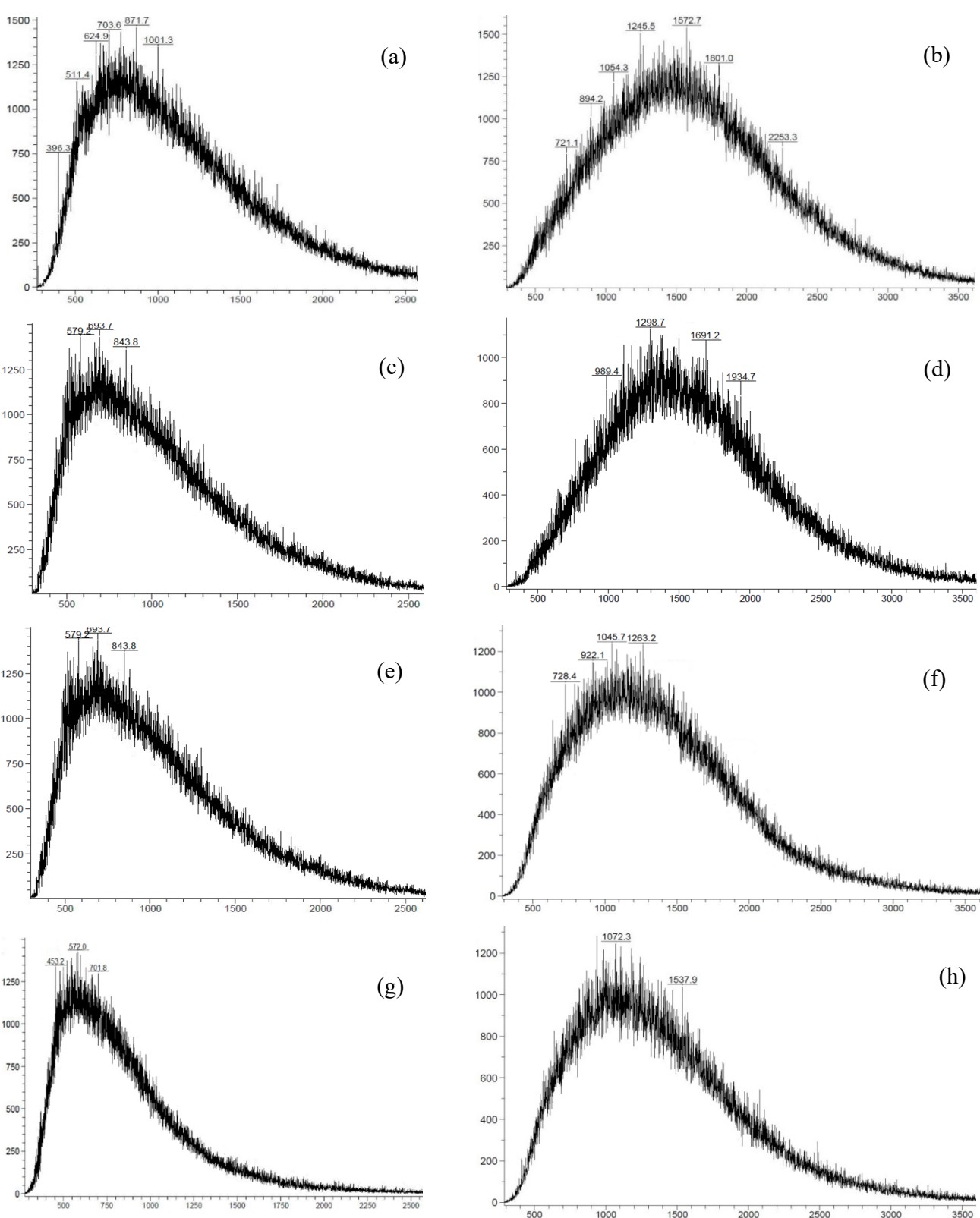

**Figure 9.** Matrix-activated laser desorption/ionization (MALDI) spectra of resins (**a**,**c**,**e**,**g**) and asphaltenes (**b**,**d**,**f**,**h**) before (**a**,**b**) and after catalytic upgrading for 48 h (**c**,**d**), 72 h (**e**,**f**), and 96 h (**g**,**h**).

## 3. Research Methods

### 3.1. Materials

In the present study, we used extra-heavy oil and oil-saturated crushed cores from the same Boca de Jaruco (Cuba) carbonate reservoir rocks as in [6,15,37], which were provided by JSC Zarubezhneft (Moscow, Russia). The group composition and elemental analysis of core extracts are summarized in Table 6. The distilled tall oil with acid number = 182 mg KOH/g and $NiSO_4 \cdot 7H_2O$ were used to synthesize the oil-soluble catalyst precursors.

**Table 6.** Group composition and elemental analysis of oil saturated crushed cores.

| Sample | SARA Fractions, wt.% | | | |
|---|---|---|---|---|
| | Saturates | Aromatics | Resins | Asphaltenes |
| | 16.5 | 31.8 | 26.4 | 25.3 |
| **Core extracts** | Elemental analysis, wt.% | | | |
| | C | H | N | S | O | H/C |
| | 83.91 | 9.09 | 0.38 | 2.65 | 3.97 | 1.301 |

### 3.2. Catalytic and Non-Catalytic Aquathermolysis Modeling in Batch Reactor Coupled with Gas Chromatography

Catalytic and non-catalytic aquathermolysis of extra-heavy oil and oil-saturated crushed cores were carried out in a high-pressure batch reactor (Parr Instruments, Moline, IL, USA). The mass ratio of bituminous crushed cores and water was 10:1. The introduced concentration of catalytic complex (4 wt.%) was calculated basing on the content of bitumoid in crushed cores, which is equal to 13.5 wt.% The set pressure and temperature in reactor was $90 \pm 3$ bar and $300 \pm 5$ °C, respectively. The different aquathermolysis reaction time (48–96 h) was tested. The short time (1–12 h) catalytic aquathermolysis experiments with extra-heavy crude oil were carried out to reveal only the mechanism of in-situ transformation of catalyst precursor. The gas products of catalytic and non-catalytic aquathermolysis were analyzed in Gas Chromatography (GC) Chromatec-Crystal 5000 (Chromatec, Yoshkar-Ola, Russia) which was coupled to batch reactor. The upgraded liquid products of crushed cores were systematically evaluated using physical and chemical instrumental analysis methods.

### 3.3. Transformation of Catalyst Precursors

Apart from catalytic and non-catalytic aquathermolysis modeling in batch reactor, series of autoclave experiments with high concentration of catalyst precursor (10 wt.%) in extra-heavy oil were conducted for relatively short-time period (1–12 h) in order to reveal the proposed transformation mechanism of catalyst precursors. The transformation products of catalyst precursors were isolated and then studied by X-ray diffraction, SEM, and EDX-mapping analysis methods.

#### 3.3.1. Isolation of Nickel-Based Catalyst

The catalyst nanoparticles were isolated from the oil and water phases by centrifuging in Eppendorf 5804R (Eppendorf, Hamburg, Germany) with a speed of 5000 rpm for 15 min and then washing organics with toluene. This operation was cycled for several times until the organic part was totally removed and the toluene solution became transparent. Then, we evaporated the toluene in drying cabinet and collected the isolated particles for further analysis.

#### 3.3.2. X-ray Diffraction Analysis

The X-ray diffraction analysis was accomplished using Shimadzu XRD-7000S automatic powder diffractometer (Shimadzu, Kyoto, Japan) using a nickel monochromator

with a step of 0.008 nm and 3 s point exposure, in combination with a Bruker D2 PHaser (Bruker, Billerica, MA, USA) and CuKα radiation with a wavelength of λ = 1.54060 nm [19].

### 3.3.3. Scanning Electron Microscope (SEM) Analysis

SEM analysis was carried out on field emission scanning electron microscope Merlin (Carl Zeiss, Oberkochen, Germany) equipped with an Energy-Dispersive X-ray spectrometer (EDX mapping) Aztec X-Max (Oxford Instruments, Abingdon, UK) to study the morphology and elemental composition of catalyst particles. Prior to microscope, samples were transferred to carbon scotch, loaded in a chamber, where a vacuum was created. Surface morphology was analyzed on secondary electron mode, the resolution of which is very high in terms of surface morphology. Elemental analysis was performed under 20 keV with angle selective backscatter mode to reveal the difference in composition.

### 3.4. Products of Catalytic and Non-Catalytic Aquathermolysis

### 3.4.1. SARA-Analysis

The core extracts before and after the aquathermolysis treatment in the presence and absence of catalytic complex were separated into four fractions: saturates, aromatics, resins, and asphaltenes according to the standards of American Society for Testing and Materials (ASTM) D2007. The oil-hexane ratio of 1:40 is suggested to precipitate asphaltenes. Hexane was chosen to keep the light fractions in maltenes. Further, maltenes were separated into saturates in liquid-adsorption chromatography filled with aluminum oxide, previously calcined at 420 °C and eluted with hexane. Then, aromatic compounds were eluted with toluene, and finally resins, which were eluted from the aluminum oxide with mixture of benzene and isopropyl alcohol (1:1).

### 3.4.2. Gas Chromatography-Mass Spectroscopy (GC-MS)

Saturates and aromatics fractions of core extracts before and after the catalytic and non-catalytic aquathermolysis were analyzed on the GC-MS system, which included the gas GC Chromatech-Crystal 5000 (Chromatech, Yoshkar-Ola, Russia) with a mass-selective detector ISQ (Thermo Fisher Scientific, Waltham, MA, USA). The Xcalibur (X-calibur Construction Systems Inc., Fort Lauderdale, FL, USA) application have been used for processing the results. A capillary column with 30 m long and a diameter of 0.25 mm was used to carry out the experiment. The carrier gas was a helium and its flow rate was equal to 1 mL/min. The set temperature was 310 °C. The adjusted thermostat regime was as follows: a rise from 100 °C to 150 °C with a speed of 3 °C/min, from 150 °C to 300 °C with a speed of 12 °C/min followed by its isotherm to the end of the analysis. Electron energy—70 eV, temperature of ion source—250 °C. The detailed compounds of saturates and aromatics fractions have been identified using NIST Mass Spectral Library and literature sources.

### 3.4.3. Fourier Transform Infrared Spectral (FT-IR) Analysis

Vector 22 IR spectrometer (Bruker, Karlsruhe, Germany) was used to analyze the changes in structural group composition of core extracts after catalytic aquathermolysis. The spectra were measured in the range of 600–1800 cm$^{-1}$. Vakhin et al. described the details of measurement and processing spectra in their recent study [38]. In this study, the following spectrometric indices were used: *$C_1$ = $D_{1600}/D_{720}$ (aromaticity), *$C_2$ = $D_{1710}/D_{1465}$ (oxidation), *$C_3$ = $D_{1380}/D_{1465}$ (branching), *$C_4$ = $(D_{720} + D_{1380})/D_{1600}$ (aliphatic), and *$C_5$ = $D_{1030}/D_{1465}$ (sulfurization).

### 3.4.4. Elemental Analysis

The elemental composition of core extracts before and after treatment was measured by X-ray fluorescence method, in M4 Tornado (Bruker, Billerica, MA, USA). The given analysis provides the data about the content of C, H, N, O, and S in core extracts before and after the catalytic treatment.

### 3.4.5. Matrix-Activated Laser Desorption/Ionization (MALDI) Analysis

UltraFlex III TOF/TOF (Bruker, Karlsruhe, Germany) was utilized to analyze the molecular mass of resins and asphaltenes fractions isolated by SARA-analysis method before and after catalytic aquathermolysis processes.

## 4. Conclusions

In this paper, we studied the upgrading performance and possible reaction paths of nickel-based oil soluble tallate for in-situ aquathermolysis of extra-heavy oil of Boca de Jaruco field. Decomposition of nickel tallate was observed by several methods during the aquathermolysis processes for 1–12 h. The transformation of nickel tallate into active form was submitted by nonstoichiometric forms of nickel sulfide such as $NiS_2$, $Ni_9S_8$, $Ni_3S_4$, $Ni_3S_2$. According to SEM images, the size of catalyst particles is in the range of 80–100 nm. It was observed that the dispersity of catalyst particles is independent from the duration of aquathermolysis processes. The catalytic performance of nickel-based nanodispersed particles in upgrading extra-heavy oil at 300 °C was investigated in high-pressure batch reactor for 48, 72 and 96 h. The influence of catalytic complex was mainly manifested in destruction of C–S bonds of resins and asphaltenes. The content of saturated hydrocarbons was increased by two times after 24 h catalytic upgrading processes. The results of SARA analysis and GC-MS spectra of aromatic fractions after catalytic upgrading showed the increase in the content of aromatic hydrocarbons, which saturated with high molecular weight polycyclic aromatic compounds such as isomers of benzo(a)fluorine—the destruction products of resins and asphaltenes. FT-IR spectra showed a slight decrease in aromaticity index ($C_1$) with increasing aliphatic indices ($C_4$). Sulfurization index ($C_5$) was decreased with time due to reduction of sulfoxides to sulfides and hydrogen sulfides. According to elemental analysis results, the catalytic upgrading mainly influenced heavy fractions of extra-heavy oil. The influence of catalytic complex on increasing of hydrogen to carbon ratio with duration of catalytic upgrading process was revealed. The average molecular mass of resins and asphaltenes after catalytic treatment was reduced from 871.7 to 523.3 a.m.u. and from 1572.7 to 1072.3 a.m.u., respectively. A study of some aspects of in-situ transformation of nickel tallate and catalytic conversion of high-molecular hydrocarbons in the presence of hydrogen donors makes such catalytic complex a promising promoter of aquathermolysis reactions that significantly improves the quality of extra-heavy oil.

**Author Contributions:** Conceptualization, A.V.V.; methodology, S.A.S.; investigation, I.I.M. and O.V.P.; data curation, I.S.A.; writing—original draft preparation, S.I.K. and I.S.A.; writing—review and editing, F.A.A. and D.K.N. All authors have read and agreed to the published version of the manuscript.

**Funding:** The authors declare no competing financial interest.

**Institutional Review Board Statement:** Not applicable.

**Informed Consent Statement:** Not applicable.

**Acknowledgments:** This work was supported by the Ministry of Science and Higher Education of the Russian Federation under agreement No. 075-15-2020-931 within the framework of the development program for a world-class Research Center "Efficient development of the global liquid hydrocarbon reserves".

**Conflicts of Interest:** The authors declare no conflict of interest.

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
