# Peer review of "Extra-Heavy Oil Aquathermolysis Using Nickel-Based Catalyst: Some Aspects of In-Situ Transformation of Catalyst Precursor"

_catalysts, doi:10.3390/catal11020189_

Round 1
Reviewer 1 Report
Vakhin and et al. in the manuscript entitled “Extra-heavy Oil Aquathermolysis Using Nickel-based Catalyst: Some Aspects of in-situ Transformation of Catalyst Precursor” describe the using Ni-based catalyst for aquathermolysis of extra-heavy oil. Although the manuscript is well organized and lays clearly the experiments performed, some details need to be addressed:
1) The entire manuscript should be edited to avoid typing, grammar, and punctuation mistakes.
2) It is unclear how much of the synthetic (and other) work on the compounds that are reported in this manuscript is novel.
3) In the section of “2.1. Object of Research”: the aim of the work was not clear.
4) In the experimental section, it is necessary to bring the procedure of the isolated Ni-based catalysts used in this research.
5) Although the results and their interpretation are well presented in the manuscript, in my opinion, it is better to use some references for some explanation like FT-IR and GC-MS analysis! Phytochemistry Letters 4 (3), 254-258, 2011.
6) Discussion of results should be in deeper form and clearly supported by recent research. Please explain the difference in your results with other research in this area!
7) The format of references should be re-checked as some of them are not match the journal’s style.
8) The author should revise minor English mistakes such as spelling error to attain high readability.
Author Response
RESPONSE TO REVIEWER 1:
We highly appreciate the reviewers’ insightful and helpful comments on our manuscript. We have spell-checked the entire manuscript and corrected some typing, grammar and punctuation mistakes. We have revised the manuscript according to the peer-review requirements. Particularly, we have rewrote the abstract section totally and replaced the «Research methods» section with «Results and Discussions» according to reviewers’ request.
Comment 1) The entire manuscript should be edited to avoid typing, grammar, and punctuation mistakes.
Response: some typing, grammar and punctuation mistakes were corrected throughout the manuscript.
Comment 2) It is unclear how much of the synthetic (and other) work on the compounds that are reported in this manuscript is novel.
Response: as far as we understood, the reviewer wants to understand the novelty of the research. If so, we would like to attract his/her attention that this particular research is the continuous of our previous researches that were already mentioned, cited and the results of which were compared in the main body of our manuscript. In this regard, tall oil, which is a by-product of wood pulp processing, was used for the first time as a ligand for synthesis of aquathermolysis catalyst precursor. Moreover, the experimental results affirming the in-situ transformation of catalyst precursor during aquathermolysis process is a novel. In addition, new data regarding the long-duration (up to 96 hours) in-reservoir catalytic aquathermolysis of extra-heavy oil were achieved.
Comment 3) In the section of “2.1. Object of Research”: the aim of the work was not clear.
Response: Thank you for pointing this. We are totally agree with you, the object of the research should reflect the aim of the research, but what we mean was the used materials and subject of the research. Therefore, we changed the title of the section to the «Materials».
Comment 4) In the experimental section, it is necessary to bring the procedure of the isolated Ni-based catalysts used in this research.
Response: The isolation procedure of nickel-based catalyst is added to the experimental section (3.3.1.).
Comment 5) Although the results and their interpretation are well presented in the manuscript, in my opinion, it is better to use some references for some explanation like FT-IR and GC-MS analysis! Phytochemistry Letters 4 (3), 254-258, 2011.
Response: we really appreciate your help regarding using some references while explaining the GC-MS and other instrumental analysis. We supported discussion of GC-MS and MALD specters by some references according to your provided article sample.
Comment 6) Discussion of results should be in deeper form and clearly supported by recent research. Please explain the difference in your results with other research in this area!
Response: The similar processes and reactions are already discussed and cited in introduction part. Regarding the «difference in your results with other research in this area» I have to remind the reviewer that crude oil is a very complex mixture, the composition and structure of crude oil depends from one reservoir to another, even from one well to another, moreover from one well depth to another well depth. Therefore, it is not appropriate to compare the catalytic upgrading products of two different oils. In addition, in literature there are only few articles on the aquathermolysis of Boca de Jaruco heavy oil to compare with. However, we cited them and compared our results with those researches. Besides, we would like to remind that this manuscript mainly focuses on some aspects of in-situ transformation of catalyst precursors, which is poorly studied. To sum up, discussion of results are detailed and supported by recent researches.
Comment 7) The format of references should be re-checked as some of them are not match the journal’s style.
Response: We appreciate the time and effort that you have provided. In this work, we used «Mendeley» to ease our work, but some references were indeed out of journal’s format. We have corrected them. Thank you.
Comment 8) The author should revise minor English mistakes such as spelling error to attain high readability
We have spell-checked the entire manuscript and corrected some typing, grammar and punctuation mistakes.

Reviewer 2 Report
This manuscript describes the aquathermolysis of extra-heavy oil over Ni-based catalyst. I recommend this manuscript be accepted for this journal after revising the following point.
- The authors tried to analyze the reaction product using various analysis methods such as SARA, GC-MS, EA, and so on. In general, the degree of reaction is determined by analyzing the boiling point distribution of the reaction product. It seems necessary to supplement this point.
- In this study, water was used as a hydrogen source to upgrade heavy oil. However, there is no result of water content change in the reaction product. Explanation of changes in water content before and after the reaction is required.
Author Response
RESPONSE TO REVIEWER 2:
We highly appreciate the reviewers’ brief but helpful comments on our manuscript. We have spell-checked the entire manuscript and corrected some typing, grammar and punctuation mistakes. We have revised the manuscript according to the peer-review requirements. Particularly, we have rewrote the abstract section totally and replaced the «Research methods» section with «Results and Discussions» according to reviewers’ request.
Comment 1. The authors tried to analyze the reaction product using various analysis methods such as SARA, GC-MS, EA, and so on. In general, the degree of reaction is determined by analyzing the boiling point distribution of the reaction product. It seems necessary to supplement this point.
Response: We are totally agree that analyzing the boiling point distribution of catalytic products is important in such upgrading processes. However, we were not able to carry this experiment due to insufficient content of bitumoid samples obtained after each catalytic upgrading products. We would like to remind dear reviewer that the subject of our study was native core samples with ≈ 13% oil saturation, which were crushed mechanically. However, we appreciate the reviewer for pointing this out and hope we will consider the boiling point distribution in the second part of our study, which will be carried out with extra-heavy crude oil instead of crushed core samples.
Comment 2. In this study, water was used as a hydrogen source to upgrade heavy oil. However, there is no result of water content change in the reaction product. Explanation of changes in water content before and after the reaction is required.
Response: We are very thankful that you pointed out this issue, as we forgot to explain the role of additionally introduced hydrogen donor – «Solvent». We now revised the introduction part by explaining the role of additional hydrogen donor and hope that it will increase the readability of our manuscript. Thus, regarding the mass balance of water we have to mention that the water is not the sole source of hydrogen donor in the given system and therefore it is not easy to provide the mass balance. As far as we know, the only way to investigate this aspect is via deuterium tracing study. The hydrogen donating capacity of water was not the purpose of our study and that is why we did not investigate this issue. However, we mentioned this phenomenon in introduction part basing on the recent researches of other scholars (we cited the researches properly). In this manuscript the focus was done on some aspects of in-situ transformation of precursors.

Round 2
Reviewer 2 Report
The authors tried to answer the reviewer’s questions. I recommend this manuscript be accepted for this journal after revising the following point.
- The authors said that hydrogen donor solvent was used for the aquathermolysis of heavy oil. However, there is no information about the solvent in the experimental section. Please make this point clearly.
Author Response
Thank you for this suggestion. We have accordingly clarified the used hydrogen donor after your previous insightful comment. We think this point was avoided by reviewer after the revision as it was mentioned in introduction section and therefore we highlighted with red color. However, we didn't provide detail information because in the case of our study, it seems slightly out of scope. Instead, we cited to our earlier studies, where the composition of the given solvent was thoroughly studied.
